# Non-rigid Relative Placement through 3D Dense Diffusion

**Eric Cai, Octavian Donca, Ben Eisner, David Held**
Robotics Institute, School of Computer Science
Carnegie Mellon University, United States
{eycai, odonca, baeisner, dheld}@andrew.cmu.edu

**Abstract:** The task of "relative placement" is to predict the placement of one object in relation to another, e.g. placing a mug onto a mug rack. Through explicit object-centric geometric reasoning, recent methods for relative placement have made tremendous progress towards data-efficient learning for robot manipulation while generalizing to unseen task variations. However, they have yet to represent deformable transformations, despite the ubiquity of non-rigid bodies in real world settings. As a first step towards bridging this gap, we propose "cross-displacement" - an extension of the principles of relative placement to geometric relationships between deformable objects - and present a novel vision-based method to learn cross-displacement through dense diffusion. To this end, we demonstrate our method's ability to generalize to unseen object instances, out-of-distribution scene configurations, and multimodal goals on multiple highly deformable tasks (both in simulation and in the real world) beyond the scope of prior works. Supplementary information and videos can be found at our website.

**Keywords:** Deformable, Non-rigid, Manipulation, Relative Placement

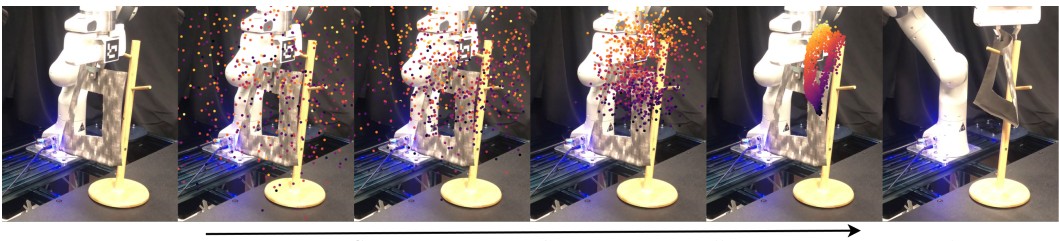

**Initial Observation** — **Goal Prediction through Cross-Displacement Diffusion** → **Successful Placement**

Figure 1: Our method (TAX3D) uses dense displacement diffusion to determine how to perform a deformable cloth-hanging task for an unseen scene configuration.

## 1 Introduction

Learning from demonstrations has emerged as a powerful technique to imbue robots with complex manipulation capabilities. Recent approaches have achieved remarkable success on a wide variety of tasks, including mug-hanging [1, 2, 3], book-shelving [4], cloth-folding [5], and sauce-pouring [6]. One common approach for imitation learning is to learn end-to-end visuomotor policies that map directly from observations to low-level robot actions; however, such approaches often struggle to generalize to novel variations of the scene (e.g. unseen object instances or object configurations). In this work, we aim to train robots to be robust to these variations for *multimodal, deformable* object placement tasks. Given demonstrations of a cloth-hanging task, for example, a robot should be able to successfully hang the cloth for unseen cloth instances or hanger positions. Similarly, if the task has multiple possible goal configurations (e.g. multiple holes on the cloth through which to orient the hanger), the robot should be able to output multimodal predictions to complete the task.

8th Conference on Robot Learning (CoRL 2024), Munich, Germany.

A simple solution is to collect an increasingly large dataset with all of the variations that one wishes to be robust to. However, collecting such a dataset, particularly in the real world, may be prohibitively infeasible. Other recent works have taken an alternative approach by reasoning more explicitly about object geometry and casting the manipulation task as a "relative placement" problem [1, 2, 3, 4]. These approaches directly model the pose relationship between the object being manipulated (the "action object") and the rest of the scene (the "anchor") in an object-centric fashion by separately featurizing action and anchor geometries; these approaches have shown the ability to learn precise manipulation tasks with impressive generalization capabilities from a small number (e.g. 10) of demonstrations [1]. However, these approaches also make strong assumptions about object rigidity, predicting an $SE(3)$ transformation that does not apply to deformable objects.

In this paper, we present **TAX3D** (**TA**sk-Specific **Cross-D**isplacement through **D**ense **D**iffusion), a deep 3D-vision framework to handle multimodal goal prediction for *deformable* relative placement tasks (e.g. hanging a cloth on a hanger). Namely, our contributions include:

(1) A formal definition of "cross-displacement," in which we extend the idea of relative placement beyond rigid poses to arbitrary deformable settings through a dense representation.
(2) A novel method to predict cross-displacement through object-centric point-wise diffusion.
(3) A novel task benchmark for multimodal relative placement for deformable object manipulation from demonstrations, building on DEDO [7].

We design experiments to rigorously evaluate performance under goal multimodality, out-of-distribution scene configurations, and variations in object geometry. Contrary to prior relative placement methods, we demonstrate that our method is robust to all of these variations for simulated and real-world cloth-hanging tasks, paving the way for generalizable non-rigid relative placement.

## 2 Related Work

**Relative Placement for Object Manipulation:** As previously described, there exists a corpus of work that deconstructs a placement task as a relative pose prediction problem [8, 9, 10, 11, 12]. Dense Object Nets (DON) [3] and Neural Descriptor Fields (NDF) [2] accomplish this by learning dense embeddings and matching observation embeddings to demonstration embeddings. While these approaches work in constrained placement settings, they assume that the target object is moved relative to a static reference object. TAX-Pose [1] addresses this issue by directly modeling the relative pose of objects in the scene and regressing a task-specific "cross-pose" from learned dense embeddings. This inference pipeline, however, renders TAX-Pose unsuitable for multimodal goal prediction. Recent work (TAX-PoseD [13]) extends TAX-Pose to multimodal tasks using a conditional Variational AutoEncoder (cVAE) with a dense spatially-grounded latent space. Relational Pose Diffusion (RPDiff) [4] leverages iterative pose de-noising to learn object-scene relationships that are multimodal, but diffuses directly in the space of $SE(3)$ transformations. This fundamentally restricts RPDiff to rigid placement tasks. In contrast to these approaches, our method is able to handle both multi-modal placements and deformable objects for relative placement tasks.

**Diffusion Models for Point Cloud Generation:** Diffusion models [14, 15, 16, 17, 18, 19, 20] have emerged as the state-of-the-art in 2D image generation; their variational loss allows for more stable training and mode capture (in contrast to GANs), and the lack of aggressive regularization on their latent distribution mitigates the coverage-fidelity tradeoff inherent to VAEs. Following this success, there have been many efforts to transfer these advantages to 3D point cloud generation, either by applying 2D diffusion techniques to volumetric representations [21, 22] or by denoising individual points independently [23, 24]. Our methodology aligns with the latter, as we adapt the Diffusion Transformer [25] architecture to de-noise pointwise displacements.

**Deformable Object Manipulation:** Despite its prevalence in the real world, deformable manipulation remains a challenge in robotics due to the complexity of its underlying dynamics and state spaces. Early works have demonstrated success on deformable tasks such as cloth and bedding

manipulation [26, 27, 28, 29, 30, 31, 32], rope manipulation [33], dough rolling [34, 35], and bag opening [36, 37]. These methods, however, often rely on pre-defined features (e.g. corner or wrinkle detection) [38, 32, 30], manually designed action primitives (e.g. pick-and-place, fold, or fling) [27, 28, 26, 38, 33, 31, 29, 34], or simplifying assumptions about the scene (e.g. a 2D tabletop environment) [31, 37, 36, 32, 35], limiting their generalization to a broad range of deformable tasks. More recently, end-to-end visuomotor policies [6, 39, 5, 40] have shown some capacity to learn non-rigid manipulation tasks. However, our experiments show that one such method (DP3 [39]) fails to generalize to novel objects or unseen object configurations, whereas our approach of non-rigid relative placement shows significantly improved generalization abilities.

## 3 Problem Statement

**Relative Placement for Rigid Objects:** In this paper, we study "relative placement" as a general framework that encapsulates many robotics tasks (e.g. placing a mug on a rack, or hanging a cloth on a hanger). Given two objects $\mathcal{A}$ and $\mathcal{B}$, the goal of a relative placement task is to manipulate object $\mathcal{A}$ (the "action" object) into some position relative to object $\mathcal{B}$ (the "anchor" object).

We will briefly review relative placement tasks for rigid objects as defined in prior work [1]: for objects $\mathcal{A}$ and $\mathcal{B}$, let point clouds $\mathbf{P}_{\mathcal{A}}^*$ and $\mathbf{P}_{\mathcal{B}}^*$ denote their respective object geometries in a desired goal configuration. Suppose, for example, that object $\mathcal{A}$ is a mug, object $\mathcal{B}$ is a mug rack, and ($\mathbf{P}_{\mathcal{A}}^*$, $\mathbf{P}_{\mathcal{B}}^*$) are the point clouds of these objects when the mug is hanging on the rack. For a relative placement task, if both objects are transformed by the same $SE(3)$-transformation $\mathbf{T}$, then the resulting configuration should also be considered a successful task completion; for example, if the mug and rack are transformed together, then the mug is still hanging on the rack. Formally, if ($\mathbf{P}_{\mathcal{A}}$, $\mathbf{P}_{\mathcal{B}}$) denote the point clouds of objects $\mathcal{A}$ and $\mathcal{B}$ in some arbitrary configuration, we can define whether this configuration represents a successful relative placement as:

$$\text{RelPlace}(\mathbf{P}_{\mathcal{A}}, \mathbf{P}_{\mathcal{B}}) = \textbf{SUCCESS} \iff \exists \mathbf{T} \in SE(3) \text{ s.t. } \mathbf{P}_{\mathcal{A}} = \mathbf{T} \cdot \mathbf{P}_{\mathcal{A}}^* \text{ and } \mathbf{P}_{\mathcal{B}} = \mathbf{T} \cdot \mathbf{P}_{\mathcal{B}}^*. \quad (1)$$

Then, given observed object point clouds $\mathbf{P}_{\mathcal{A}}$ and $\mathbf{P}_{\mathcal{B}}$, the goal of a rigid relative placement task [1] is to learn a function $f(\mathbf{P}_{\mathcal{A}}, \mathbf{P}_{\mathcal{B}}) = \mathbf{T}_{AB}$ that predicts a rigid transformation of object $\mathcal{A}$ such that $\text{RelPlace}(\mathbf{T}_{AB} \cdot \mathbf{P}_{\mathcal{A}}, \mathbf{P}_{\mathcal{B}}) = \textbf{SUCCESS}$. The transform $\mathbf{T}_{AB}$ is referred to as the "cross-pose" [1] since it captures the pose *relationship* between objects $\mathcal{A}$ and $\mathcal{B}$. Using motion planning, the robot can then move object $\mathcal{A}$ into a new pose determined by $\mathbf{T}_{AB}$ to complete the relative placement task, e.g. to place the mug onto the rack [1].

**Cross-Displacement for Deformables:** In the deformable case, however, this formulation is ill-defined. For a task with deformable object $\mathcal{A}$, there is no guarantee that there exists a rigid transformation $\mathbf{T}_{AB}$ that can transform an observed point cloud $\mathbf{P}_{\mathcal{A}}$ into a goal configuration $\mathbf{P}_{\mathcal{A}}^*$ such that $\text{RelPlace}(\mathbf{T}_{AB} \cdot \mathbf{P}_{\mathcal{A}}, \mathbf{P}_{\mathcal{B}}) = \textbf{SUCCESS}$, as object $\mathcal{A}$ may need to undergo a non-rigid transformation to achieve its goal configuration. Consider, for example, the task of hanging a towel on a towel rack - the deformations undergone by the towel as it folds and drapes over the hanger cannot be captured by a rigid transformation.

Instead, to model object deformations, we learn a dense transformation function that predicts where each *point* in object $\mathcal{A}$ must move. Namely, given an initial observation of object $\mathcal{A}$ as point cloud $\mathbf{P}_{\mathcal{A}} = \{x_0, \ldots, x_N\}$ with $x_i \in \mathbb{R}^3$, we aim to learn a function $g(\mathbf{P}_{\mathcal{A}}, \mathbf{P}_{\mathcal{B}}) = \Delta X$, where $\Delta X = \{\Delta x_0, \ldots, \Delta x_N\}$ with $\Delta x_i \in \mathbb{R}^3$, such that $\text{RelPlace}(\mathbf{P}_{\mathcal{A}} + \Delta X, \mathbf{P}_{\mathcal{B}}) = \textbf{SUCCESS}$. Intuitively, $\Delta X$ is a set of dense displacements such that $x_i + \Delta x_i$ brings each point $x_i$ to the goal configuration relative to $\mathbf{P}_{\mathcal{B}}$. As such, we refer to $\Delta X$ as the **"cross-displacement"** of objects $\mathcal{A}$ and $\mathcal{B}$, analogous to the "cross-pose" defined previously.

**Goal Multimodality:** For some tasks, there may be more than one way to achieve a goal configuration. For example, for the task of hanging a towel on a rack, if object $\mathcal{B}$ consists of multiple towel racks, then hanging the towel on any of the racks should yield a valid goal configuration. To this end, we can define a *distributional* relative placement task: for a given point cloud $\mathbf{P}_{\mathcal{B}}^*$ of object $\mathcal{B}$,

let $m(\mathbf{P}_\mathcal{B}^*)$ be a set of point clouds for object $\mathcal{A}$ such that for all $\mathbf{P}_\mathcal{A}^* \in m(\mathbf{P}_\mathcal{B}^*)$, the configuration pair $(\mathbf{P}_\mathcal{A}^*, \mathbf{P}_\mathcal{B}^*)$ completes the task. We can then explicitly define success for a distributional relative placement task as:

$$\text{RelPlace}_D(\mathbf{P}_\mathcal{A}, \mathbf{P}_\mathcal{B}) = \textbf{SUCCESS} \iff \exists \mathbf{T} \in SE(3), \mathbf{P}_\mathcal{A}^* \in m(\mathbf{P}_\mathcal{B}^*)$$
$$\text{s.t. } \mathbf{P}_\mathcal{A} = \mathbf{T} \cdot \mathbf{P}_\mathcal{A}^* \text{ and } \mathbf{P}_\mathcal{B} = \mathbf{T} \cdot \mathbf{P}_\mathcal{B}^*. \tag{2}$$

This definition extends the idea of distributional relative placement defined in prior work [13] to a continuous rather than discrete set of possible goal configurations. To achieve a multi-modal deformable relative placement task, we aim to learn a distribution over cross-displacements $g(\mathbf{P}_\mathcal{A}, \mathbf{P}_\mathcal{B}) = p(\Delta \mathbf{X})$ from which we can sample a candidate cross-displacement $\Delta X \sim p(\Delta \mathbf{X})$.

**Assumptions:** For our method, we assume access to segmentations for action object $\mathcal{A}$ and anchor object $\mathcal{B}$. During training, we assume access to a set of $M$ demonstration configurations that complete the task $\{(\mathbf{P}_{\mathcal{A},1}, \mathbf{P}_{\mathcal{A},1}^*, \mathbf{P}_{\mathcal{B},1}^*), \ldots, (\mathbf{P}_{\mathcal{A},M}, \mathbf{P}_{\mathcal{A},M}^*, \mathbf{P}_{\mathcal{B},M}^*)\}$, where $(\mathbf{P}_{\mathcal{A},i}, \mathbf{P}_{\mathcal{A},i}^*, \mathbf{P}_{\mathcal{B},i}^*)$ respectively denote the initial action object point cloud, goal action object point cloud, and goal anchor object point cloud in demonstration $i$. We also assume access to correspondences between the initial $\mathbf{P}_{\mathcal{A},i}$ and the goal $\mathbf{P}_{\mathcal{A},i}^*$ point clouds; these correspondences are required to compute the ground-truth cross-displacement $\Delta X^* = \mathbf{P}_\mathcal{A}^* - \mathbf{P}_\mathcal{A}$ used to supervise our model as explained below.

# 4 Method

Given point clouds $(\mathbf{P}_\mathcal{A}, \mathbf{P}_\mathcal{B})$ of objects $\mathcal{A}$ and $\mathcal{B}$ in an arbitrary configuration, our goal is to learn a function that predicts a distribution over cross-displacements $g(\mathbf{P}_\mathcal{A}, \mathbf{P}_\mathcal{B}) = p(\Delta \mathbf{X})$. From this distribution, we can sample a cross-displacement $\Delta X \sim p(\Delta \mathbf{X})$ to transform $\mathbf{P}_\mathcal{A}$ into a successful goal configuration $\hat{\mathbf{P}}_\mathcal{A} = \mathbf{P}_\mathcal{A} + \Delta X$ relative to object $\mathcal{B}$.

To solve this distributional non-rigid relative placement problem, we leverage diffusion models [14, 15], a class of generative models that rely on iterative noising and de-noising. More specifically, our method builds on Improved Denoising Diffusion Probabilistic Models [41]. Given some Gaussian noise $x_T$ (noised from the forward process [14]), we train our model to de-noise $x_T$ by learning the reverse diffusion process [14]: $p_\theta(x_{t-1} \mid x_t) = \mathcal{N}(x_{t-1}; \mu_\theta(x_t), \Sigma_\theta(x_t))$. Following [41], we re-parameterize the mean $\mu_\theta(x_t)$ and the covariance $\Sigma_\theta(x_t)$ using a noise prediction $\epsilon_\theta(x_t)$ and interpolation vector $v_\theta(x_t)$, respectively, and supervise with a hybrid loss function that combines the standard noise prediction error with a variational lower bound loss; for further details, we defer to Nichol and Dhariwal [41].

## 4.1 Point Cloud Generation for Relative Placement

**Training:** Given a sample demonstration $(\mathbf{P}_\mathcal{A}, \mathbf{P}_\mathcal{A}^*, \mathbf{P}_\mathcal{B}^*)$, where $\mathbf{P}_\mathcal{A}$ is the initial action object point cloud $\{x_{\mathcal{A},0}, \ldots, x_{\mathcal{A},N}\}$, $\mathbf{P}_\mathcal{A}^*$ is the goal action object point cloud $\{x_{\mathcal{A},0}^*, \ldots, x_{\mathcal{A},N}^*\}$, and $\mathbf{P}_\mathcal{B}^*$ is the goal anchor object point cloud, we train our model to de-noise a set of per-point displacements $\Delta X$, with the ground-truth displacements defined as $\Delta X^* = \mathbf{P}_\mathcal{A}^* - \mathbf{P}_\mathcal{A} = \{x_{\mathcal{A},0}^* - x_{\mathcal{A},0}, \ldots, x_{\mathcal{A},N}^* - x_{\mathcal{A},N}\}$. Following Ho et al. [15], we sample partially noised displacements $\Delta X_t$:

$$\Delta X_t = \sqrt{\bar{\alpha}_t}\Delta X_0 + \sqrt{1 - \bar{\alpha}_t}\epsilon \tag{3}$$

where $\Delta X_0 = \Delta X^*$, $t \sim [0,T]$, $\epsilon \sim \mathcal{N}(\mathbf{0}, \mathbf{I})$, and $\bar{\alpha}_t$ is a function of the noise schedule. We then supervise our model's noise $\epsilon_\theta(\Delta X_t, \mathbf{P}_\mathcal{A}, \mathbf{P}_\mathcal{B}^*, t)$ and interpolation vector $v_\theta(\Delta X_t, \mathbf{P}_\mathcal{A}, \mathbf{P}_\mathcal{B}^*, t)$ predictions using the hybrid loss function from Nichol and Dhariwal [41]. Note that the model is conditioned on the anchor $\mathbf{P}_\mathcal{B}^*$, as well as the initial action point cloud $\mathbf{P}_\mathcal{A}$, which we refer to below as the "action context." For architecture and training specifics, see Appendix B.

**Inference:** At inference, given some object configuration $(\mathbf{P}_\mathcal{A}, \mathbf{P}_\mathcal{B})$, we initialize a set of per-point displacements as Gaussian noise: $\Delta X_T \sim \mathcal{N}(0, \mathbf{I})$. These displacements are iteratively de-noised using our model's outputs: $\Delta X_{t-1} \sim \mathcal{N}(\Delta X_{t-1}; \mu_\theta, \Sigma_\theta)$, with $\mu_\theta$ and $\Sigma_\theta$ re-parameterized as in Nichol and Dhariwal [41]. The final de-noised displacements $\Delta X_0$ are then used to transform the points of $\mathbf{P}_\mathcal{A}$ into a predicted placement $\mathbf{P}_\mathcal{A} + \Delta X_0$ relative to $\mathbf{P}_\mathcal{B}$.

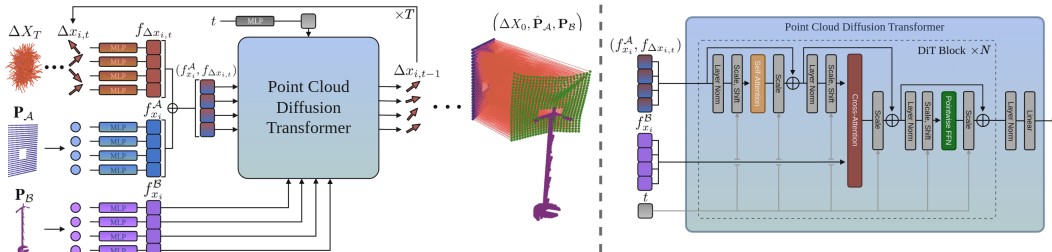

Figure 2: *(Left)*. During inference, randomly sampled displacements $\Delta X_T \sim \mathcal{N}(0, \mathbf{I})$ are de-noised conditioned on action ($\mathbf{P}_{\mathcal{A}}$) and anchor ($\mathbf{P}_{\mathcal{B}}$) features; the final $\Delta X_0$ is predicted to displace the action into a goal configuration. *(Right)*. Our modified DiT [25] architecture combines self-attention and cross-attention for object-centric and scene-level reasoning.

**Object-Specific Frames:** To guarantee translation-invariance in our model, we process input point clouds ($\mathbf{P}_{\mathcal{A}}, \mathbf{P}_{\mathcal{B}}$) in object-specific frames. Namely, we center the action context $\mathbf{P}_{\mathcal{A}}$ in the action frame ($\mathbf{P}'_{\mathcal{A}} = \mathbf{P}_{\mathcal{A}} - \bar{\mathbf{P}}_{\mathcal{A}}$) and the anchor $\mathbf{P}_{\mathcal{B}}$ in the anchor frame ($\mathbf{P}'_{\mathcal{B}} = \mathbf{P}_{\mathcal{B}} - \bar{\mathbf{P}}_{\mathcal{B}}$) by subtracting their respective point cloud means $\bar{\mathbf{P}}_{\mathcal{A}}$ and $\bar{\mathbf{P}}_{\mathcal{B}}$. Accordingly, our model is conditioned on ($\mathbf{P}'_{\mathcal{A}}, \mathbf{P}'_{\mathcal{B}}$) instead of ($\mathbf{P}_{\mathcal{A}}, \mathbf{P}_{\mathcal{B}}$). During training, we further center the ground-truth goal action point cloud in the anchor frame ($\mathbf{P}^{*'}_{\mathcal{A}} = \mathbf{P}^{*}_{\mathcal{A}} - \bar{\mathbf{P}}_{\mathcal{B}}$) and modify the ground-truth displacements to $\Delta X^{*'} = \mathbf{P}^{*'}_{\mathcal{A}} - \mathbf{P}'_{\mathcal{A}}$. During inference, we can easily transform our predictions back into the world frame by inverting these translations. In our experiments, we show that this use of object-specific frames is crucial for robust out-of-distribution generalization.

## 4.2 Diffusion Transformer for Point Cloud Generation

We adapt the Diffusion Transformer (DiT) [25] architecture for our model, as shown in Figure 2. At each diffusion timestep, our model takes as input the noised displacements $\Delta X_t$, the action context $\mathbf{P}_{\mathcal{A}}$, the anchor $\mathbf{P}_{\mathcal{B}}$, and the timestep $t$. Using MLP encoders, we first compute per-point displacement features $f_{\Delta X}$, per-point action context features $f_x^{\mathcal{A}}$, and per-point anchor features $f_x^{\mathcal{B}}$ from $\Delta X_t$, $\mathbf{P}_{\mathcal{A}}$, and $\mathbf{P}_{\mathcal{B}}$, respectively. MLP weights are shared within sets of features. The action context features and displacement features are then concatenated into $(f_x^{\mathcal{A}}, f_{\Delta x})$ as input to our modified DiT model alongside anchor features $f_x^{\mathcal{B}}$ (see Figure 2, left).

Within a DiT block, self-attention is applied to the combined action features $(f_{x_i}^{\mathcal{A}}, f_{\Delta x_i})$ to aggregate information across the entire action point cloud and facilitate coordinated displacement predictions. Cross-attention is then applied to these features with the anchor features $f_x^{\mathcal{B}}$, allowing for scene-level global reasoning between the action and anchor objects. This is repeated for $N$ blocks, before the network outputs a noise $\epsilon_\theta$ and interpolation vector $v_\theta$ prediction, as described above.

We refer to the above approach as the **"Cross-Displacement (CD)"** variant of our TAX3D architecture, since we directly predict the cross-displacement $\Delta X$. We also propose a **"Cross-Point (CP)"** variant in which we directly encode and diffuse over the positions of the predicted goal point cloud $\hat{\mathbf{P}}_{\mathcal{A}} = \mathbf{P}_{\mathcal{A}} + \Delta X$ instead; we find that both variants perform similarly in our experiments. Following Peebles and Xie [25], we incorporate timestep conditioning using adaptive layer normalization (adaLN) blocks. For permutation equivariance, we do not incorporate any positional encoding.

## 5 Experiments

**Setup:** To the best of our knowledge, quantitative benchmarks do not exist for the setting of deformable object relative placement. As such, to evaluate our approach, we design a novel experimental benchmark building on DEDO: Dynamic Environments with Deformable Objects [7], a suite of simulation environments for deformable manipulation. We experiment on two cloth-hanging tasks (Figure 3): `HangProcCloth`, in which a cloth must be hung through one of its holes, and `HangBag`,

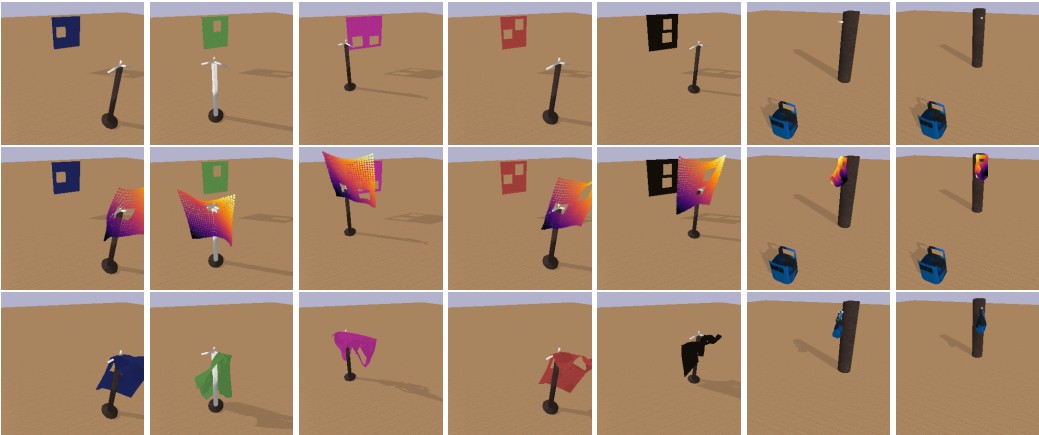

Figure 3: TAX3D generalizes to diverse cloths and anchor positions (*top*); we also visualize the corresponding goal predictions (*middle*) and successful rollouts (*bottom*) after releasing the cloth. The two rightmost columns are `HangBag` configurations - all others are `HangProcCloth` configurations.

in which a bag must be hung on one of its handles. For environment and demonstration details, see Appendix A. We also evaluate our method in the real world, described further below.

**Evaluation Metrics:** We evaluate our approach using two metrics: point prediction error and downstream policy performance. To measure point prediction error, we use point-wise **root-mean-square-error (RMSE)** to compute the distance between a prediction $\hat{\mathbf{P}}_{\mathcal{A}}$ and a ground truth goal configuration $\mathbf{P}_{\mathcal{A}}^*$. To evaluate *distributional* placements for multimodal experiments, we also report **Coverage RMSE**, in which we compute the minimum distance to a set of sampled predictions $\{\hat{\mathbf{P}}_{\mathcal{A},j}\}$ for each ground truth goal $\mathbf{P}_{\mathcal{A},i}^*$, and **Precision RMSE**, in which we compute the minimum distance to the set of ground truth goals $\{\mathbf{P}_{\mathcal{A},i}^*\}$ for each sampled prediction $\hat{\mathbf{P}}_{\mathcal{A},j}$. To evaluate downstream policy performance, we use a primitive goal-conditioned policy that directly converts a predicted goal point cloud $\hat{\mathbf{P}}_{\mathcal{A}}$ into position control inputs and report the cloth hanging **Success Rate**. Further details regarding evaluation can be found in Appendix C.

**Ablations and Baselines:** We evaluate the two variants of our method described in Section 4.2 - **"Cross-Displacement" (CD)** and **"Cross-Point" (CP)** - against several architectural ablations:

(1) **Scene Displacement/Point (SD/SP):** a variant of our CD/CP architectures *without object-centric reasoning*. The action and anchor point clouds are combined into a single scene point cloud $\mathbf{P}_{\mathcal{S}}$, and the DiT blocks are implemented with self-attention only.
(2) **Cross Displacement/Point - World Frame (CD-W/CP-W):** a variant of our CD/CP architectures *without object-specific frames*: $\mathbf{P}_{\mathcal{A}}$ and $\mathbf{P}_{\mathcal{B}}$ are not mean-centered.
(3) **Cross Displacement/Point - No Action Context (CD-NAC/CP-NAC):** a variant of our CD/CP architectures *without action context features*. $\mathbf{P}_{\mathcal{A}}$ is not encoded, but cross-attention between displacement features $f_{\Delta x}$ and anchor features $f_x^{\mathcal{B}}$ is retained.

We also compare our method to **3D Diffusion Policy (DP3)** [39], a recent end-to-end visuomotor policy that has achieved state-of-the-art manipulation results using point cloud inputs.

**Generalization to Unseen Configurations and Geometries:** Experiments are conducted on two versions of the `HangProcCloth` task (`unimodal`, in which the cloth has one hole, and `multimodal`, in which the cloth has two holes) as well as on the `HangBag` task. Important comparisons are shown in Tables 1, 3, and 4, respectively, with full results in Appendix D. Training demonstrations are generated under random anchor poses (64 demonstrations for `HangProcCloth` and 16 for `HangBag`), and models are evaluated on two sets of configurations (40 trials each): **Unseen**, which contains novel anchor poses from the training distribution, and **Unseen (Out-of-Distribution)**, which contains out-of-distribution anchor poses. For both `HangProcCloth` tasks, cloth and hole shape are

|  | RMSE (↓) | | | Success Rate (↑) | | |
|---|---|---|---|---|---|---|
|  | **Train** | **Unseen** | **Unseen (OOD)** | **Train** | **Unseen** | **Unseen (OOD)** |
| SD | 0.127 | 0.805 | 4.246 | 0.27 | 0.20 | 0.00 |
| SP | 0.095 | 0.779 | 3.955 | 0.45 | 0.48 | 0.08 |
| CD-W | 0.099 | 0.464 | 12.464 | 0.95 | 0.88 | 0.00 |
| CP-W | 0.085 | 0.511 | 19.923 | 0.95 | 0.85 | 0.00 |
| CD-NAC | 2.233 | 2.243 | 2.326 | 0.16 | 0.28 | 0.10 |
| CP-NAC | 3.987 | 3.934 | 3.923 | 0.02 | 0.00 | 0.03 |
| *TAX3D-CD (Ours)* | 0.052 | 0.405 | **0.395** | 0.94 | 0.90 | **0.85** |
| *TAX3D-CP (Ours)* | **0.051** | **0.390** | 0.422 | 0.93 | **0.95** | 0.80 |
| DP3 | - | - | - | **0.98** | 0.38 | 0.03 |

Table 1: `HangProcCloth-unimodal`: Random Cloth Geometry (1-Hole). We do not report RMSEs for DP3 since it is an end-to-end policy and does not predict point clouds.

randomized across demonstrations, with **Unseen** and **Unseen (OOD)** consisting only of unseen cloth geometries. For the `HangBag` task, only a single fixed cloth geometry is used.

|  | Success Rate (↑) | |
|---|---|---|
|  | **Unseen** | **Unseen (OOD)** |
| *TAX3D-CD* | **1.00** | **0.98** |
| *TAX3D-CP* | **1.00** | **0.98** |
| DP3 | 0.93 | 0.00 |

Table 2: `HangProcCloth-simple`: Fixed Cloth Geometry

All ablations (Table 1) fail on out-of-distribution anchor poses, demonstrating that our method's combination of *object-level* reasoning in *object-specific* frames is crucial for generalization to novel scene configurations. As the point prediction errors indicate, NAC (No Action Context) ablations are unable to produce meaningful cloth geometries (Appendix D) due to the lack of action context - crucially, without combined action features ($f_x^{\mathcal{A}}, f_{\Delta x}$), the network cannot maintain correspondences between displacements $\Delta x_i$ and action points $x_i$.

We also find that DP3 fails to generalize both to novel scene configurations *and* to novel object instances, despite fitting well to training demonstrations. To understand why, we conduct an additional experiment with *fixed cloth geometry* (`HangProcCloth-simple`, Table 2), in which DP3 achieves decent performance on **Unseen** (but in-distribution) anchor poses. This indicates that the poor performance on **Unseen** in `HangProcCloth-unimodal` is a result of DP3's inability to adapt to *variations in the cloth geometry*. TAX3D, on the other hand, generalizes well to both scene and object variations, maintaining significantly superior performance across all `HangProcCloth` and `HangBag` experiments (Tables 1, 3, 4) in comparison to DP3.

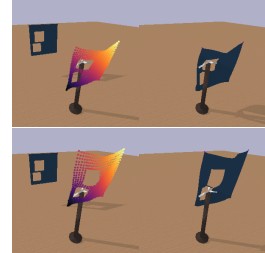

Figure 4: Multimodal TAX3D predictions (*left*), with successful rollouts (*right*).

**Multimodal Goal Prediction:** Our diffusion-based architecture naturally accommodates multimodal settings (Figure 4). To explicitly measure this, we additionally compare against regression baselines (**Regression Displacement/Point (RD/RP)**) for `HangProcCloth-multimodal`. To do so, we remove the timestep conditioning from the architecture and train the model to directly predict $\hat{\mathbf{P}}_{\mathcal{A}}$ via a regression loss (MSE). As Table 3 indicates, the regression models lead to poor performance on the multimodal task variant of a cloth with two holes. In contrast, our method continues to achieve good performance on this task; the low coverage and precision RMSEs indicate that it can predict both modes, while avoiding the mode-averaging tendencies of regressive inference.

**Real World Experiments:** We also qualitatively demonstrate our approach in a real-world cloth-hanging setup. Human demonstration videos are captured using a single Azure Kinect camera. Segmentations and correspondences are then obtained using point-prompted Segment Anything [42]

|  | Coverage RMSE (↓) | | Precision RMSE (↓) | | Success Rate (↑) | |
|---|---|---|---|---|---|---|
|  | Unseen | Unseen (OOD) | Unseen | Unseen (OOD) | Unseen | Unseen (OOD) |
| RD | 2.987 | 3.057 | 1.087 | 1.114 | 0.60 | 0.50 |
| RP | 1.417 | 1.432 | 1.192 | 1.154 | 0.65 | 0.48 |
| *TAX3D-CD (Ours)* | 0.457 | 0.543 | **0.403** | **0.558** | **0.98** | **0.73** |
| *TAX3D-CP (Ours)* | **0.453** | **0.540** | 0.495 | 0.631 | 0.85 | 0.70 |
| DP3 | - | - | - | - | 0.45 | 0.00 |

Table 3: `HangProcCloth-multimodal`: Random Cloth Geometry (2-Hole)

|  | RMSE (↓) | | | Success Rate (↑) | | |
|---|---|---|---|---|---|---|
|  | Train | Unseen | Unseen (OOD) | Train | Unseen | Unseen (OOD) |
| *TAX3D-CD (Ours)* | 0.183 | 0.204 | 0.245 | 0.94 | 0.83 | **0.78** |
| *TAX3D-CP (Ours)* | **0.173** | **0.192** | **0.233** | 0.94 | **0.93** | **0.78** |
| DP3 | - | - | - | **0.98** | 0.38 | 0.03 |

Table 4: `HangBag`: Fixed Cloth Geometry

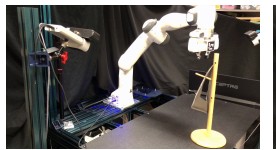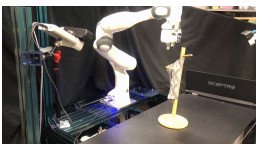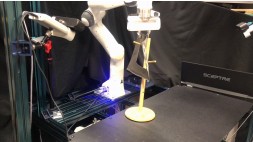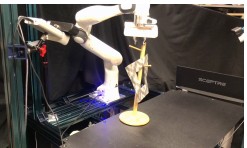

Figure 5: Real world results. TAX3D succeeds under varying anchor poses, varying peg placements (*left, middle-left*), and can model multimodal placements with multiple pegs (*middle-right, right*).

and SpatialTracker [43], respectively. We train our model on 10 demonstrations, containing multimodal placements and various anchor poses. To perform robot actions, we use position control to track the predicted goal point cloud. As shown in Figure 5, our method can complete the task under novel anchor poses and anchor geometries, while accommodating multimodal goal predictions.

## 6 Conclusion

In this paper, we present a novel framework to perform relative placement for non-rigid manipulation tasks. We formulate the task of "cross-displacement" prediction to handle relative placement for arbitrary object deformations, and we show that our dense diffusion architecture can learn cross-displacements on multiple cloth hanging tasks in simulation and in the real world. Our experiments demonstrate our approach's ability to generalize to out-of-distribution scene configurations, unseen object instances, and multimodal placements for robust deformable manipulation.

**Limitations:**    There are several limitations to our method, which we leave for future work:

1. **Segmentations.** Our method requires segmented action and anchor point clouds. Our current approach for obtaining such segmentations (point-prompted Segment Anything [42]) requires human involvement, limiting scalability. This can potentially be addressed using language-conditioned segmentation methods [44] to automate demonstration labeling.
2. **Open-loop control.** Our method currently relies on open-loop position control to the predicted goal for robot execution, which limits the ability to perform more complex placement tasks. This can potentially be addressed by incorporating TAX3D goal predictions into a goal-conditioned manipulation policy.

## Acknowledgments

This material is based upon work supported by the Toyota Research Institute, the National Science Foundation under NSF CAREER Grant No. IIS-2046491, and NIST under Grant No. 70NANB23H178. We are grateful to Prof. Shubham Tulsiani for his valuable feedback and discussion at various stages of the project, and toLifan Yu for her assistance with real-world experiments.

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

# Appendix

## Table of Contents

# A  DEDO Environment Details

DEDO: Dynamic Environments with Deformable Object [7] is a suite of task-based simulation environments (hanging a bag, dressing a mannequin, etc.) involving highly deformable, topologically non-trivial objects. The environments are built on the PyBullet physics engine [45].

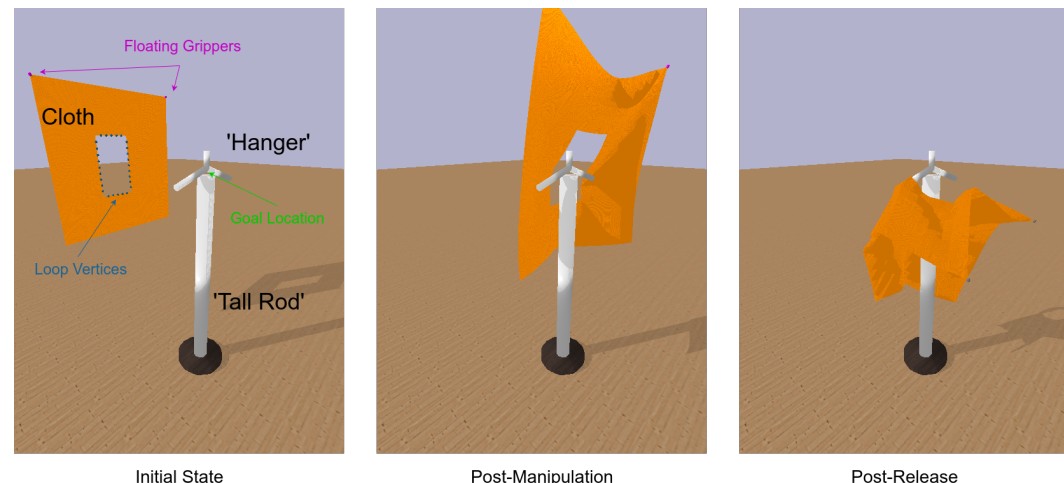

Figure 6: Sample demonstration of the `HangProcCloth` task.

## A.1  `HangProcCloth` - Task Definition

For most of our experiments, we focus on the `HangProcCloth` task (Figure 6), in which a procedurally generated cloth must be placed on a hanger. More specifically, the cloth is generated to contain a hole in its topology - to successfully complete the task, the vertical part of the hanger should be aligned *through* the hole.

The hanger (anchor) is loaded into the PyBullet engine as a pre-defined rigid body, and contains two components: a 'tall rod', and the 'hanger' itself. While we randomize the anchor pose throughout our experiments, this geometry remains fixed. The **goal** of the task is explicitly formulated in the environment as the center of the 'hanger' component (Figure 6) - while this goal definition is not passed as input to our models, it is used later by our success metric for evaluation.

## A.2  `HangProcCloth` - Cloth Generation

Following DEDO's implementation, every cloth in our experiments is procedurally generated as a rectangular mesh, and can be represented using the following parameters:

| | | |
|---|---|---|
| node_density | 25 | The amount of vertices to initialize the cloth mesh with. Every cloth is initialized as an evenly spaced `node_density` × `node_density` grid (25 ×25 = 625 vertices for all of our cloths). Vertices are then removed during the hole generation process. |
| width | [0.8, 1.2] | The width of the cloth. |
| height | [0.8, 1.2] | The height of the cloth. |
| num_holes | (1..2) | The number of holes in the cloth. |
| holes | See A.2.1 | See A.2.1 |

### A.2.1 `HangProcCloth` - Hole Generation

Holes are created by removing mesh vertices. All generated holes are rectangular - as such, they can be represented topologically with respect to the procedurally generated cloth by their bottom-left and top-right corners. Accordingly, the `holes` parameter is a list, where each element corresponding to a specific hole in the cloth is a dictionary with elements:

| | |
|---|---|
| x0 | The $x$ vertex coordinate of the bottom-left corner of the hole. |
| y0 | The $y$ vertex coordinate of the bottom-left corner of the hole. |
| x1 | Similar to x0, for the top-right corner. |
| y1 | Similar to y0, for the top-right corner. |

For reference, the single-hole cloth used in our `HangProcCloth-simple` experiment is defined as:

```
1  {
2      "node_density": 25,
3      "width": 1.0,
4      "height": 1.0,
5      "num_holes": 1,
6      "holes": [
7          {"x0": 8, "y0": 9, "x1": 16, "y1": 13}
8      ]
9  }
```

In general, holes are randomly generated under the following constraints:

| | | |
|---|---|---|
| x_range | (2, node_density - 2) | The range of possible values for x0. |
| y_range | (2, node_density - 2) | The range of possible values for y0. |
| width_range | (5, 7) | The range of possible values for $w_h$, such that x1 = x0 + $w_h$. |
| height_range | (5, 7) | The range of possible values for $w_h$, such that x1 = x0 + $w_h$. |

More precisely, when generating holes, x0 and y0 are first sampled based on x_range and y_range, respectively - x1 and y1 are then sampled based on width_range and height_range. To ensure that the resulting cloth geometry is valid topologically, DEDO generates cloths using a Monte Carlo method, only returning valid holes if they pass a boundary check (all vertices lie within the cloth boundary) and an overlap check (different holes do not overlap). For further implementation details, we refer to the DEDO codebase [7].

Since holes are generated by directly manipulating the cloth mesh, they can also be represented as deformable loops, defined by a set of "loop vertices" (Figure 6) - while information about these vertices is not passed as input to our models, it is used later by our success metric for evaluation.

### A.3 `HangBag` - Task Definition

We also perform an additional experiment on the `HangBag` task (Figure 7), in which a deformable bag with two handles must be placed on a hanger. To successfully complete the task, at least one of the handles must lie on the handle.

Similar to `HangProcCloth`, the hanger (anchor) is a pre-defined rigid body - for our experiments, we randomize the anchor pose, but keep the geometry fixed. Unlike `HangProcCloth`, however, the cloth is loaded from a pre-defined mesh rather than procedurally generated. As such, we do not randomize the cloth geometry for this task, leaving such experiments for future work (notably, the DEDO environment does provide multiple meshes for the bag).

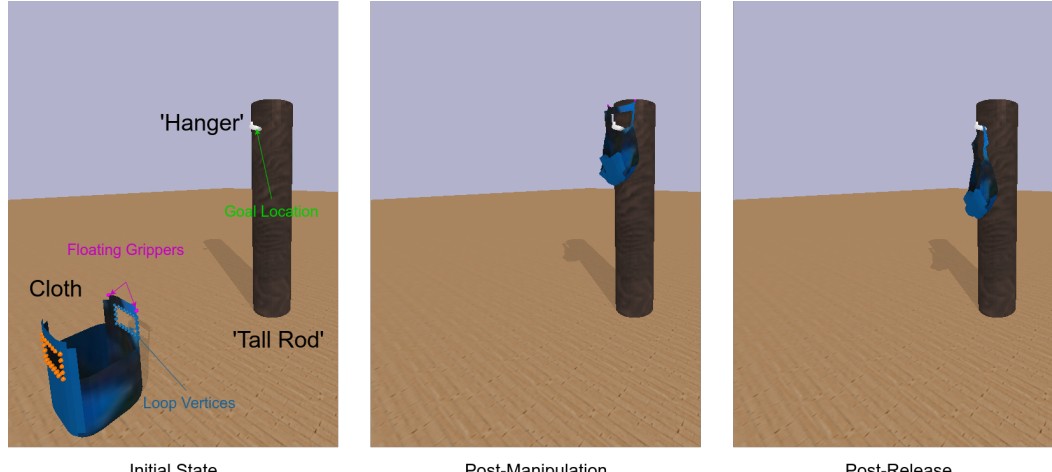



Initial State       Post-Manipulation       Post-Release

Figure 7: Sample demonstration of the `HangBag` task.



## A.4 Cloth Control

The `HangProcCloth` and `HangBag` environments do not model a robot grasp - instead, the cloth is manipulated by applying force controls to floating "grippers"[1] attached to pre-defined points on the cloth (Figure 6,7). The grippers themselves are zero-mass and collision free.

**Pseudo-expert Policy:** To generate demonstrations, we hard-code a pseudo-expert policy with access to privileged environment information. At the initial state of the scene, the policy computes a target position for each gripper using the distance from the centroid of the loop vertices to the goal location in the hanger. If the cloth has multiple holes, a single hole is selected. When computing this target, we apply the same transformation that we apply to the anchor to randomize its pose - this ensures that the cloth is rotated appropriately to align with the hanger. At each time step, we also add a small correction to the target position based on the distance between the current centroid of the loop vertices and the goal position - this allows the pseudo-expert policy to adapt to small deformations in the cloth geometry, which we find meaningfully improves the success rate. For control, we pass the target positions for each gripper as inputs to a custom proportional-derivative controller (in addition to a target velocity of 0) with a velocity and position gains of 50 and a maximum force[2] of 5.

**Evaluation Policy:** To evaluate our models, we implement a separate evaluation policy without access to privileged environment information (e.g. goal location, deformable loop vertices). At the initial state of the scene, we run TAX3D on the full point cloud of the cloth[3], and obtain the predicted position in the world frame of the two grippers (which we assume are attached to known points of the cloth). These target positions are then passed as inputs (in addition to a target velocity of zero) to a custom proportional-derivative controller, with the same gains as the pseudo-expert policy.

## A.5 Episode Rollout

### A.5.1 Rollout Phases

Each episode rollout consists of two phases (Figure 6): a **manipulation phase**, in which the grippers receive force control inputs at each time step to manipulate the cloth, and a **release phase**, in which

---

[1]DEDO refers to these grippers as "anchors." We refrain from this terminology since "anchor" denotes an entirely different object for our purposes.

[2]Following the DEDO implementation, this is not an overall maximum force magnitude - it is the maximum magnitude of the force along the $x$-, $y$-, and $z$-axes.

[3]Note that this is different from our training procedure ( B.2), where we downsample cloth point clouds to 512 points.

the grippers "release" the cloth and allow it to fall. The release phase is fixed at 500 simulation steps, whereas the manipulation phase has a variable episode length depending on the setting (with each environment step corresponding to 8 simulation steps).

If the task is completed successfully, the cloth should be supported by the rigid anchor *after* the release phase. However, because we are learning a goal-prediction module to condition a policy's control outputs, we use the post-manipulation, pre-release state of the cloth to label ground truth demonstrations.

### A.5.2   Success Metric

To robustly determine whether or not the task has been successfully completed, we implement our own success metric consisting of two components:

1. **Centroid Check:** a binary metric that checks if the centroid of the deformable loop vertices is within a threshold distance of the goal location.
2. **Polygon Check:** a binary metric that projects the loop vertices and goal location onto the $xy$-plane, and then checks if the projected goal point lies on the interior of the polygon defined by the projected loop vertices. This is an intuitive heuristic that checks whether or not the hole "wraps" around the vertical rod of the hanger.

If the cloth has multiple holes, these metrics are computed individually for each hole - the task is considered successful if both are true for at least one hole.

For the `HangProcCloth` task, the success metric consists of a centroid check (with threshold 1.3) pre-release and a polygon check post-release. For the `HangBag` task, the success metric conists of two centroid checks, both with threshold 1.4, pre- and post-release - we do not use the polygon check for the `HangBag` task, as we found that it produces a larger number of false negatives.

### A.6   Demonstration Generation

### A.6.1   Randomizing Scene Configuration

For all `HangProcCloth` experiments, the objects in the scene are initialized to the following pose, shown in Figure 6:

|        | position $(xyz)$ | orientation (Euler) |
|--------|------------------|---------------------|
| cloth    | $(0, 5, 8)$ | $\left(-\frac{\pi}{2}, 0, \frac{3\pi}{2}\right)$ |
| hanger   | $(0, 0, 8)$ | $(0, 0, 0)$ |
| tall rod | $(0, 0, 0)$ | $(0, 0, 0)$ |

Table 5: Initial `HangProcCloth` configuration

For all `HangBag` experiments, the objects in the scene are initialized to the following pose instead (shown in Figure 7):

|        | position $(xyz)$ | orientation (Euler) |
|--------|------------------|---------------------|
| cloth    | $(0, 8, 2)$    | $\left(\frac{\pi}{2}, 0, 0\right)$ |
| hanger   | $(0, 1.28, 9)$ | $\left(0, 0, \frac{\pi}{2}\right)$ |
| tall rod | $(0, 0, 5)$    | $\left(\frac{\pi}{2}, 0, 0\right)$ |

Table 6: Initial `HangBag` configuration

To randomize scene configuration, the anchor is then transformed with a randomly sampled translation, and a randomly sampled rotation about the $z$-axis.

| | Unseen | Unseen (OOD) |
|---|---|---|
| $x$-translation | $(-5, 5)$ | $(-10, -5) \cup (5, 10)$ |
| $y$-translation | (0, -10) | (0, -10) |
| $z$-translation | 0 | $(1, 5)$ |
| $z$-rotation | $\left(-\frac{\pi}{3}, \frac{\pi}{3}\right)$ | $\left(-\frac{\pi}{3}, \frac{\pi}{3}\right)$ |

All transformations are sampled uniformly at random from their respective ranges, with one small caveat: $x$-translations are chosen such that their signs match the sign of the sampled rotation. That is, if the $z$-rotation is sampled to be non-negative, then the $x$-translations are only sampled from the non-negative subset of the corresponding range. This ensures that the anchor always "faces" the cloth, such that the cloth need not undergo significant rotations for a successful placement.

As for the point clouds themselves, we obtain $\mathbf{P}_{\mathcal{B}}^*$ as a partial point cloud from an RGB-D render of the initial state of the environment (since the anchor is static). To guarantee correspondences, $\mathbf{P}_{\mathcal{A}}$ and $\mathbf{P}_{\mathcal{A}}^*$ are directly extracted from the mesh vertices of the cloth at its initial and post-manipulation states, respectively.

### A.6.2 Experiment Datasets

As a reminder, cloth geometry (including holes) is randomized by following the parameter ranges and procedures described in A.2.1. The datasets for each experiment are generated as follows, where each tuple entry corresponds to the (**Train, Unseen, Unseen (OOD)**) settings, respectively:

| | # cloths | # holes per cloth | # total demonstrations |
|---|---|---|---|
| `HangProcCloth-simple` | (1, 1, 1) | (1, 1, 1) | (16, 40, 40) |
| `HangProcCloth-unimodal` | (64, 40, 40) | (1, 1, 1) | (64, 40, 40) |
| `HangProcCloth-multimodal` | (32, 20, 20) | (2, 2, 2) | (64, 40, 40) |
| `HangBag` | (1, 1, 1) | (1, 1, 1) | (16, 40, 40) |

For `HangProcCloth-multimodal`, we generate a successful demonstration for each hole per cloth - as such, there are half as many unique cloths as there are total demonstrations. For all demonstrations across all experiments, the anchor pose is randomized as described in A.6.1.

## B  Training Details

### B.1  Model Architecture

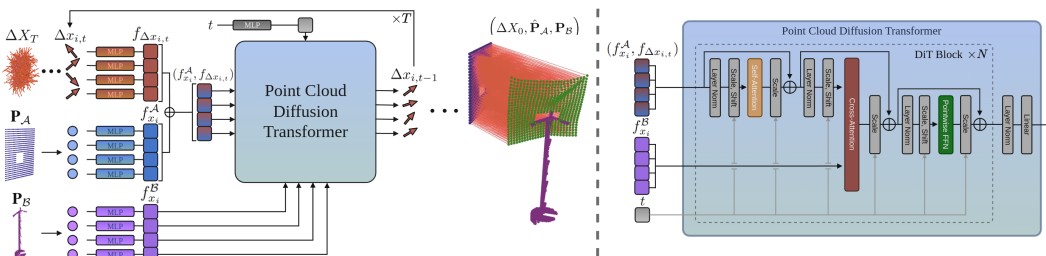

Figure 8: TAX3D model architecture. *(Left)*. During inference, randomly sampled displacements $\Delta X_T \sim \mathcal{N}(0, \mathbf{I})$ are de-noised conditioned on action ($\mathbf{P}_{\mathcal{A}}$) and anchor ($\mathbf{P}_{\mathcal{B}}$) features; the final $\Delta X_0$ is predicted to displace the action into a goal configuration. *(Right)*. Our modified DiT [25] architecture combines self-attention and cross-attention for object-centric and scene-level reasoning.

As discussed in 4.2, we modify the standard DiT block [25] to include an additional cross-attention head 8. For all of our experiments, we train the same architecture:

| | | |
|---|---|---|
| depth | 5 | # of DiT blocks |
| num_heads | 4 | # heads per block |
| hidden_size | 128 | hidden size per block |

These settings (namely, num_heads and hidden_size) are applied identically to the self-attention and cross-attention layers. During training and inference, our model always uses 100 diffusion steps, with a linear noise schedule.

## B.2 Training Pre-Processing & Hyperparameters

For training, both the action and anchor point clouds are downsampled to 512 points using furthest point sampling[4]. The anchor point cloud is additionally augmented with $z$-axis rotations sampled uniformly at random from $[0, 2\pi]$.

All models are trained under the same hyperparameters with AdamW optimization and cosine scheduling with warmup:

| | |
|---|---|
| learning_rate | $1 \times 10^{-4}$ |
| learning_rate_warmup_steps | 100 |
| weight decay | $1 \times 10^{-5}$ |
| epochs | 20,000 |
| batch_size | 16 |

# C   Evaluation Metrics

As discussed in Section 5, our method's modeling of point-wise displacements allows us to directly use root-mean-squared-error (RMSE) as a distance metric between predicted and ground truth configurations of the cloth. To appropriately evaluate distributional predictions in our setting, we define two evaluation metrics[5]:

1. **Coverage RMSE:** For each demonstration with ground truth $\mathbf{P}^*_{\mathcal{A},i}$, we sample 20 predictions $\{\hat{\mathbf{P}}_{\mathcal{A},j}\}$, and keep the minimum RMSE. This is aggregated across all demonstrations in the dataset. Intuitively, this metric captures how well a model can produce all of the modes in a given dataset - that is, how well it *covers* a distribution.

2. **Precision RMSE:** We first collect demonstrations corresponding to a specific cloth geometry (this is either one demonstration for the unimodal case, or two demonstrations for the multimodal case) - for some cloth $\mathcal{C}$, this serves as a cloth-specific reference set $\{\mathbf{P}^*_{\mathcal{A},i}\}_{\mathcal{C}}$. We then sample 20 predictions conditioned on cloth $\mathcal{C}$, and compute for each prediction $\hat{\mathbf{P}}_{\mathcal{A},j}$ the minimum RMSE to ground truth point clouds in the reference set $\{\mathbf{P}^*_{\mathcal{A},i}\}_{\mathcal{C}}$[6]. This is aggregated across all 80 predictions, and then across all cloths. Intuitively, this metric captures how well a model can consistently produce predictions that are close to the dataset configurations - that is, how *precisely* it models a distribution.

# D   Experiments

The following sections contain all of the ablation and baseline comparisons for all experiments. Note that we do not report any RMSE metrics for DP3, since it is an end-to-end policy, and does not predict a point cloud.

---

[4]During policy evaluation, only the anchor point cloud is downsampled, as the full action point cloud is need to obtain target positions for the grippers.

[5]Both metrics bear strong similarity to the MMD metric, but are essentially modified to aggregate across different reference sets.

[6]In the multimodal case, all demonstrations for a single cloth are recorded under the same anchor configuration - as such, the world frame RMSEs are consistent across any cloth-specific reference set $\{\mathbf{P}^*_{\mathcal{A},i}\}_{\mathcal{C}}$.

## D.1 `HangProcCloth-simple` Results

| | RMSE (↓) | | | Success Rate (↑) | | |
|---|---|---|---|---|---|---|
| | **Train** | **Unseen** | **Unseen (OOD)** | **Train** | **Unseen** | **Unseen (OOD)** |
| SD | 0.054 | 0.741 | 2.898 | 0.94 | 0.98 | 0.00 |
| SP | **0.051** | **0.172** | 2.969 | 0.50 | 0.58 | 0.05 |
| CD-W | 0.052 | 0.224 | 3.680 | **1.00** | **1.00** | 0.00 |
| CP-W | 0.044 | 0.231 | 3.452 | **1.00** | **1.00** | 0.00 |
| CD-NAC | 1.586 | 1.498 | 1.741 | 0.56 | 0.60 | 0.53 |
| CP-NAC | 3.563 | 3.573 | 3.534 | 0.00 | 0.00 | 0.00 |
| *TAX3D-CD (Ours)* | 0.270 | 0.255 | **0.498** | **1.00** | **1.00** | **0.98** |
| *TAX3D-CP (Ours)* | 0.298 | 0.277 | 0.518 | **1.00** | **1.00** | **0.98** |
| DP3 | - | - | - | **1.00** | 0.93 | 0.00 |

Table 7: `HangProcCloth-simple`: Fixed Cloth Geometry

## D.2 `HangProcCloth-unimodal` Results

| | RMSE (↓) | | | Success Rate (↑) | | |
|---|---|---|---|---|---|---|
| | **Train** | **Unseen** | **Unseen (OOD)** | **Train** | **Unseen** | **Unseen (OOD)** |
| SD | 0.127 | 0.805 | 4.246 | 0.27 | 0.20 | 0.00 |
| SP | 0.095 | 0.779 | 3.955 | 0.45 | 0.48 | 0.08 |
| CD-W | 0.099 | 0.464 | 12.464 | 0.95 | 0.88 | 0.00 |
| CP-W | 0.085 | 0.511 | 19.923 | 0.95 | 0.85 | 0.00 |
| CD-NAC | 2.233 | 2.243 | 2.326 | 0.16 | 0.28 | 0.10 |
| CP-NAC | 3.987 | 3.934 | 3.923 | 0.02 | 0.00 | 0.03 |
| *TAX3D-CD (Ours)* | 0.052 | 0.405 | **0.395** | 0.94 | 0.90 | **0.85** |
| *TAX3D-CP (Ours)* | **0.051** | **0.390** | 0.422 | 0.93 | **0.95** | 0.80 |
| DP3 | - | - | - | **0.98** | 0.38 | 0.03 |

Table 8: `HangProcCloth-unimodal`: Random Cloth Geometry (1-Hole)

## D.3 `HangProcCloth-multimodal` Results

| | RMSE (↓) | | | Success Rate (↑) | | |
|---|---|---|---|---|---|---|
| | **Train** | **Unseen** | **Unseen (OOD)** | **Train** | **Unseen** | **Unseen (OOD)** |
| SD | 1.048 | **1.361** | 3.832 | 0.36 | 0.48 | 0.05 |
| SP | **1.028** | 1.407 | 3.825 | 0.53 | 0.40 | 0.05 |
| CD-W | 1.436 | 1.535 | 5.006 | 0.95 | 0.63 | 0.03 |
| CP-W | 1.418 | 1.555 | 48.760 | **0.97** | 0.70 | 0.08 |
| CD-NAC | 2.467 | 2.560 | 2.650 | 0.25 | 0.10 | 0.10 |
| CP-NAC | 4.148 | 4.237 | 4.114 | 0.00 | 0.00 | 0.00 |
| RD | 2.978 | 2.987 | 3.057 | 0.53 | 0.60 | 0.50 |
| RP | 1.360 | 1.417 | **1.432** | 0.47 | 0.65 | 0.48 |
| *TAX3D-CD (Ours)* | 1.413 | 1.532 | 1.610 | 0.95 | **0.98** | **0.73** |
| *TAX3D-CP (Ours)* | 1.400 | 1.568 | 1.608 | 0.94 | 0.85 | 0.70 |
| DP3 | - | - | - | **0.97** | 0.45 | 0.00 |

Table 9: `HangProcCloth-multimodal`: Random Cloth Geometry (2-Hole)

| | Coverage RMSE (↓) | | | Precision RMSE (↓) | | |
|---|---|---|---|---|---|---|
| | **Train** | **Unseen** | **Unseen (OOD)** | **Train** | **Unseen** | **Unseen (OOD)** |
| SD | 0.075 | 0.740 | 3.500 | 1.114 | 1.379 | 4.214 |
| SP | 0.061 | 0.773 | 3.462 | 1.093 | 1.432 | 4.062 |
| CD-W | 0.054 | **0.426** | 4.269 | 0.132 | 0.477 | 4.351 |
| CP-W | 0.047 | 0.446 | 33.868 | 0.099 | 0.504 | 47.372 |
| CD-NAC | 1.484 | 1.741 | 1.812 | 1.785 | 2.011 | 2.167 |
| CP-NAC | 3.593 | 3.720 | 3.603 | 3.770 | 3.914 | 3.786 |
| RD | 2.978 | 2.987 | 3.057 | 1.296 | 1.087 | 1.114 |
| RP | 1.360 | 1.417 | 1.432 | 1.305 | 1.192 | 1.154 |
| *TAX3D-CD (Ours)* | 0.040 | 0.457 | 0.543 | **0.084** | **0.403** | **0.558** |
| *TAX3D-CP (Ours)* | **0.038** | 0.453 | **0.540** | 0.087 | 0.495 | 0.631 |
| DP3 | - | - | - | - | - | - |

Table 10: `HangProcCloth-multimodal`: Random Cloth Geometry (2-Hole)

## D.4  `HangBag` **Results**

| | RMSE (↓) | | | Success Rate (↑) | | |
|---|---|---|---|---|---|---|
| | **Train** | **Unseen** | **Unseen (OOD)** | **Train** | **Unseen** | **Unseen (OOD)** |
| SD | 0.079 | 0.346 | 2.041 | 0.00 | 0.00 | 0.00 |
| SP | 0.062 | **0.125** | 1.886 | 0.00 | 0.00 | 0.00 |
| CD-W | 0.053 | 0.178 | 3.337 | **1.00** | 0.78 | 0.00 |
| CP-W | **0.046** | 0.173 | 4.204 | **1.00** | 0.80 | 0.03 |
| CD-NAC | 1.254 | 1.301 | 1.309 | 0.56 | 0.38 | 0.33 |
| CP-NAC | 1.485 | 1.487 | 1.486 | 0.00 | 0.03 | 0.00 |
| *TAX3D-CD (Ours)* | 0.183 | 0.204 | 0.245 | 0.94 | 0.83 | **0.78** |
| *TAX3D-CP (Ours)* | 0.173 | 0.192 | **0.233** | 0.94 | **0.93** | **0.78** |
| DP3 | - | - | - | **1.00** | 0.73 | 0.00 |

Table 11: `HangBag`: Fixed Cloth Geometry

## D.5  Additional Visualizations

The following pages contain visualizations of our method (as well as all baseline methods) across classes of cloth geometry (single- and double-hole) on out-of-distribution scene configurations. For every visualized prediction, both the cloth and configuration were unseen by the model during training.

Within each table row, the top displays the result of the evaluation policy rollout on the corresponding model's predicted cloth configuration; the bottom displays the predicted configuration itself. Zooming in may be necessary to properly view the policy executions. For more visualizations, including videos of the policy rollout and the full reverse diffusion process, see our project page.

## D.5.1 Single-Hole Cloth, Unseen Configuration (Out-of-distribution)

SD | CD-W | CD-P | CD-NAC | *TAX3D-CD (Ours)* | *TAX3D-CP (Ours)*

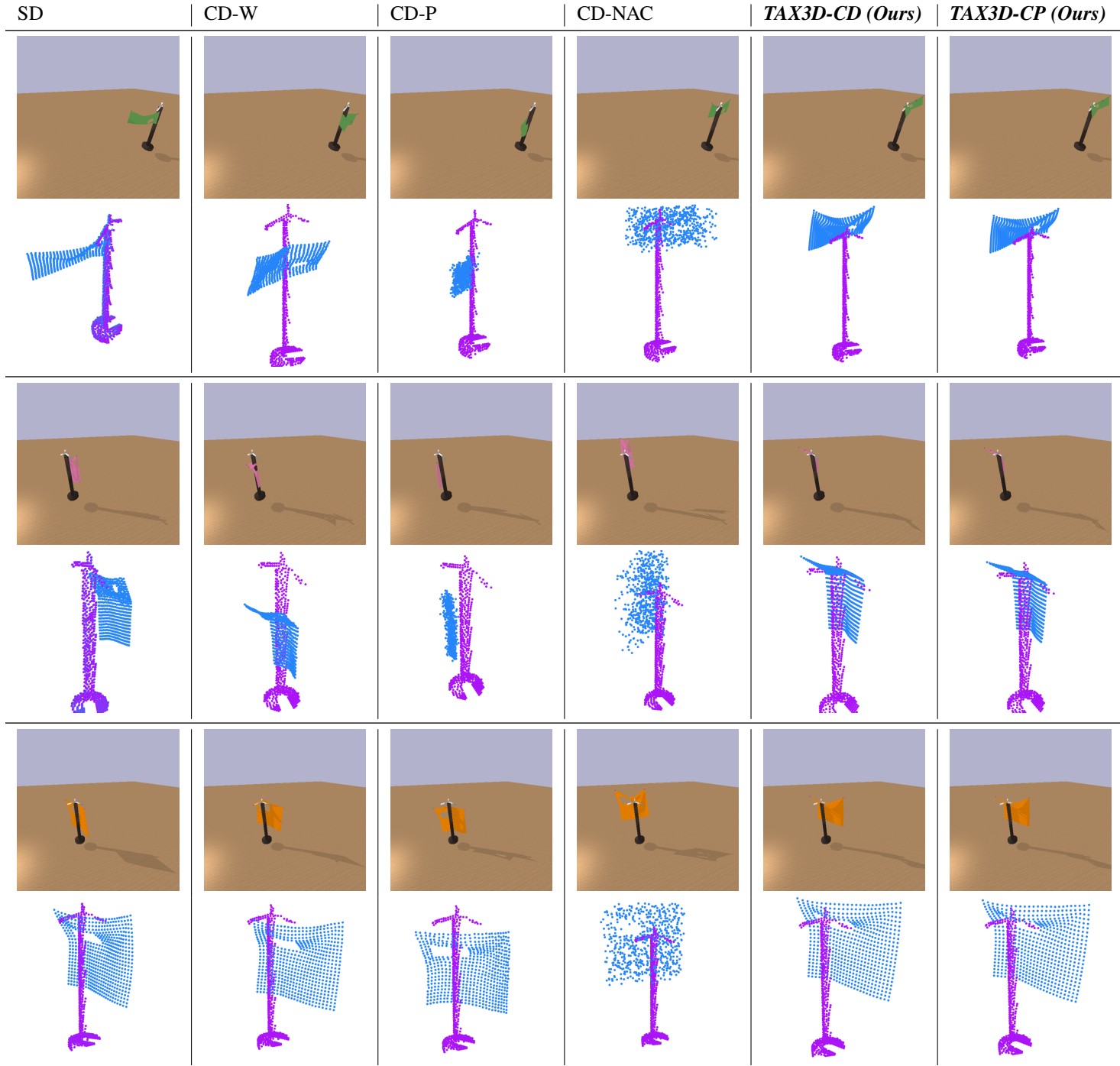

**D.5.2 Double-Hole Cloth, Unseen Configuration (Out-of-distribution)**

| SD | CD-W | CD-P | CD-NAC | *TAX3D-CD (Ours)* | *TAX3D-CP (Ours)* |

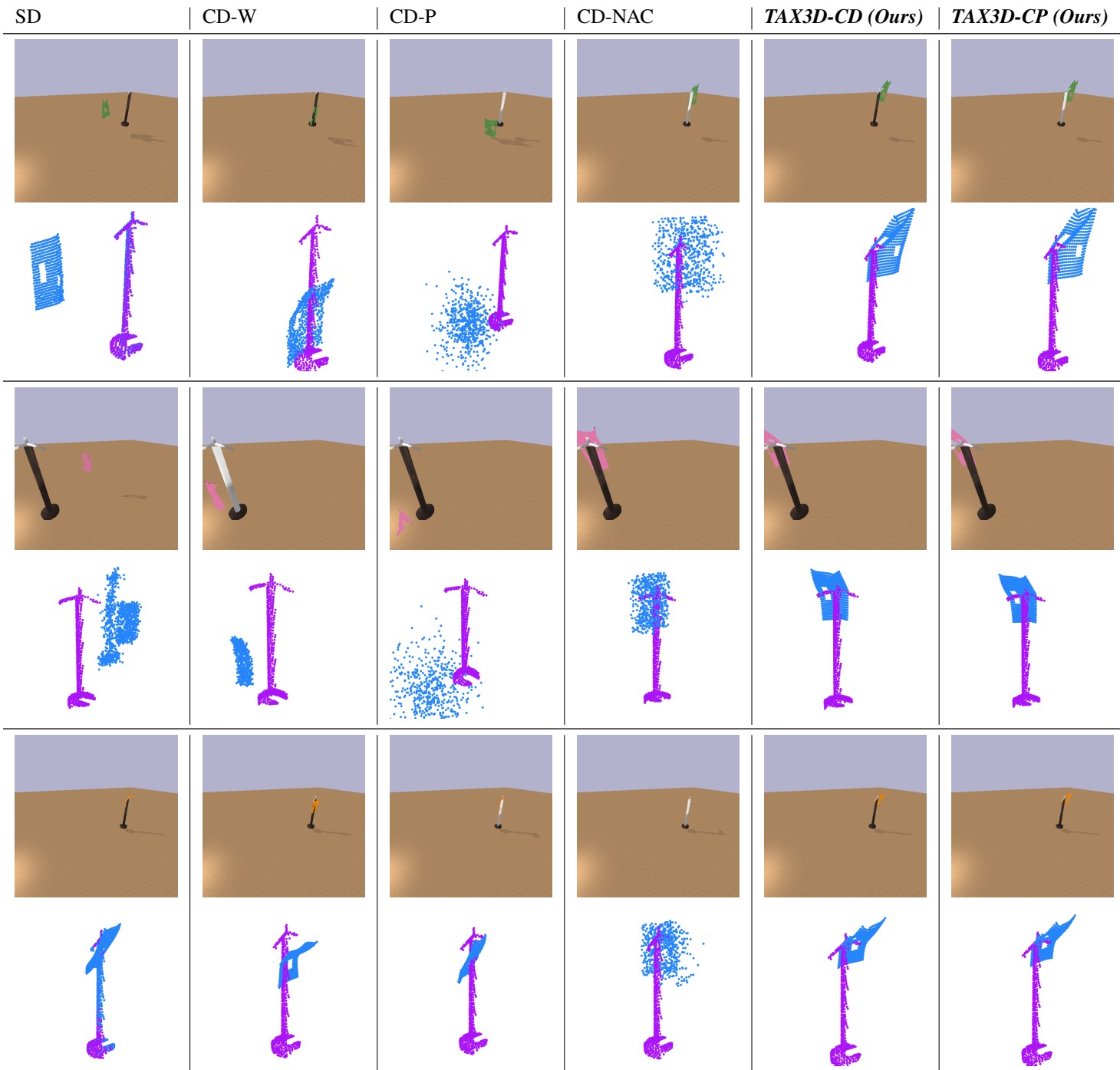

