# OpenReview forum: "Non-rigid Relative Placement through 3D Dense Diffusion"
_robot-learning.org/CoRL/2024/Conference — CoRL 2024_

### Official Review · Reviewer_6Ki8 · 2024-07-17
**Cool method, interesting paper, concerns about weak experiments**

**Originality:** 3
**Technical Quality:** 3
**Clarity Of Presentation:** 4
**Potential Impact:** 3
**Recommendation:** 2
**Confidence:** 5

**Review:**

1. Cross-displacement definition as a first contribution, re: I would argue that this is not a contribution. This term need not exist separately from relative placement as the paper continues to use the term “relative placement” in reference to deformable non-rigid objects.

2. Introduction does not contain any high level information about the method to be presented. Therefore, when reaching the related work section, it’s very difficult to understand how the method differs or compares against the related works referenced.

3. Continuation of point 1. above: it actually becomes more clear later why the distinction exists which could justify the introduction of a new term (“cross-displacement”). Suggest that a) this somehow be made clear earlier in the introduction and b) that authors use the terms consistently where they apply (i.e. relative placement when referring to rigid objects, cross-displacement when referring to non-rigid objects).

4. Ln. 137: Please expand L_{VLB} abbreviation to help the reader.

5. Ln. 164-165: P_A bar, P_B bar. Please define new notation when introduced.

6. Nit: Ln. 164-165: “anchor and goal prediction” followed by the notation in parentheses in reverse order. Please keep the order of reference as it is confusing for the reader.

7. Fig. 3: I admit I’m having a hard time understanding the cloth-hanger example shown. One of the reasons deformable dynamics are hard to learn in general is gravity. The policy rollouts do not seem to respect gravity (i.e. produce a cloth configuration that is plausible under physics constraints) so I’m unclear what the success criteria is. Then again, looking at the GT demonstrations right below, they also don’t seem to adhere to physics constraints which is doubly confusing.

8. I’m not sure how informative Coverage RMSE is as a metric. Diffusion models are multimodal and capture several distribution modes. Therefore a set of predictions would have to somehow be guaranteed to contain at least one candidate per mode that each of the goals PA*i correspond to in order to give minimum distance. Absence of such a candidate in the evaluated set may not necessarily mean poor performance of the learned model but e.g. that such a candidate didn’t happen to be sampled for the evaluated set. Thus, it seems like there are situations that the metric may not be meaningful. Vice versa for Precision RMSE. My intuition could be wrong, perhaps authors could elaborate a bit on the meaning of these metrics.

9. Ln. 217: object-centric reasoning was mentioned once in the abstract and now here. It was not really explained what that refers to in this context.

10. Fig. 4 seems cut out at the top.

11. General comment: References list is rather short.

12. General comment: Experiment list is weak. Method was evaluated on only one task in simulation.

**Quality Of The Limitations Section:**

3

**Questions For Rebuttal:**

1. Fig. 1: What is the square hole on the cloth? Typically cloths are one piece and could be hung on any spot on the fabric which would make the problem a lot more multimodal. I can’t tell why this example is multimodal if the cloth is only assumed to be hung via the existing hole.

2. How realistic are your assumptions if this method were to be applied in real settings?

3. Why center-mean the goal prediction on the anchor frame and not the action context frame? Latter seems more intuitive if you’re only predicting one Dx set to bring PA to PB. Or are you predicting multiple Dx sets in the style of trajectory waypoints in which case PB would make more sense as the anchor is stationary through the trajectory? (I don’t believe you predict multiple but just to clarify).

4. Is PA hat not the same thing as PA*? I don’t see a difference. If so, please consider using consistent notation. Edit: later on in the paper, PA hat is used for the prediction where PA* is used for the goal (so GT?) therefore in ln. 163 and 165 was the notation supposed to be PA*?

5. Ln. 179-180: Out of curiosity, did you also try just full attention over action features and anchor features as opposed to cross attention? Wondering how the results change.

6. Ln. 186-187: What are the ranges of values of PA_hat vs the range for Dx in your experiments? I’d expect that if PA_hat was significantly larger range/values than Dx, this would make some notable difference in performance.

7. How are you normalizing PA_hat and Dx in your experiments? Normalization in diffusion models tends to have a huge impact.

8. Ln. 187-188: Did you notice any difference between using adaLN blocks vs not?

9. How do you explain that Unseen OOD has a better success rate than Unseen for the 1-2 hole experiment?

**Robotics Focus:**

3

**Summary Of Paper:**

The paper leverages diffusion transformers to perform cross-displacement prediction (an extension of relative placement) between an object point cloud and an anchor point cloud.

**Summary Of Recommendation:**

Thank you to the authors. The paper is very clear and addresses a problem that is quite under researched. Also the method is cool and uses recent advancements in diffusion models that have shown to perform well in various robotics settings. The main concern which is the reason for the recommendation is the weakness of experiments. Method is evaluated only in simulation on one single task which seems to be contrived (cloth with one or two square holes being hung from a hanger without adhering to gravity). Suggest strengthening experiments in a more realistic setting and on a few different tasks involving deformables in order to demonstrate wider applicability of the method.

---

### Official Review · Reviewer_3d53 · 2024-07-25

**Originality:** 3
**Technical Quality:** 3
**Clarity Of Presentation:** 3
**Potential Impact:** 3
**Recommendation:** 3
**Confidence:** 3

**Review:**

**Strengths**
- The paper is generally very clearly written.
- It points out the issues with existing relative placement and proposes a new problem setting. While not revolutionary, it is a reasonable extension.
- The study consistently covers dataset creation, the development of the proposed method, and comparison with benchmarks.

**Weaknesses**
- The explanation of the method is insufficient. For example, according to Figure 2, while Delta X and \(P_A\) are mixed before the diffusion, \(P_B\) is mixed using cross-attention during the diffusion process. The reason for using such an architecture is unclear.
- Additionally, the description of the novelty of the method is lacking. There is a section called Point Cloud Generation for Relative Placement, but it seems that the architecture is not particularly innovative, with the main difference being the conditioned data for different problem settings.
- The relationship between the method and the issue of multimodality is unclear.
- Although a simulation dataset is provided, there is no testing in real-world environments.

**Minor**
- The link to the website is missing

**Quality Of The Limitations Section:**

2

**Questions For Rebuttal:**

See weakness section

**Robotics Focus:**

3

**Summary Of Paper:**

This study proposes a new problem called cross-displacement, which extends the existing relative placement to non-rigid objects. This extends transformations to dense transformations to accommodate non-rigid objects. To evaluate this problem setting, a new benchmark based on DENO is proposed, and the effectiveness of the proposed method based on a diffusion model is demonstrated by comparing it with several baselines.

**Summary Of Recommendation:**

I think the paper can be acceptable as in current format, but can be strengthen significantly by polishing the description about the proposed method and robotics experiments.

---

### Official Review · Reviewer_QqMb · 2024-07-31
**Review of CoRL 2024 Submission 180: Non-rigid Relative Placement through 3D Dense Diffusion**

**Originality:** 2
**Technical Quality:** 2
**Clarity Of Presentation:** 3
**Potential Impact:** 1
**Recommendation:** 2
**Confidence:** 4

**Review:**

**Strengths**

* The main method (“Point Cloud Diffusion Transformer”) is a sensible extension of Diffusion Transformers ([25] in the paper). The novelty is in the choice of inputs/outputs, the (re-)addition of cross attention, and removal of positional encoding, although the latter has already been shown in e.g. Point-E (https://arxiv.org/abs/2212.08751).
* The paper is well written and easy to follow.

**Major Weaknesses**

* Experiments are conducted entirely in simulation, and the simulation does not involve a robot at all. It is thus doubtful whether the submission is within scope of CoRL, which the website defines as: “All CoRL submissions must demonstrate the relevance to Robot Learning through
Intent: Explicitly address a learning question for physical robots OR
Outcome: Test the proposed learning solution on physical robots.” As it stands, the paper essentially presents a point-cloud warping technique only.
* The choice of project acronym (TAX3D) seems to me like at least a borderlining violation of the anonymity requirements. It is highly unlikely that authors without some overlap with authors of works in the TAX* family would choose this type of naming.
* The problem formulation (Sec. 3) is artificially narrow. There is no actual need to achieve perfect placement of every point of the object. On the contrary, it would have been important to model the fact that the placement of the majority of points is completely irrelevant to the cloth hanging task as long as the hole ends up in the right position (not even pose!).
* The narrow problem formulation probably explains why the authors haven’t found any related work. But does this respect the spirit of what scholars are expected to cover under related work/state of the art? There’s a rich body of work on robotic clothes manipulation.
* The problem formulation also focuses exclusively on a goal state. It ignores the importance of the object dynamics and its dynamic interactions with the anchor entirely. This is crucial for real-world applicability of the method.
* Similarly, the choice of a point cloud to represent the deformable object is questionable since it ignores constraints between points that would model the dynamics of just how deformable different parts of the object really are.
* The use of terminology such as “a novel vision based method” or “3D-vision” is highly misleading. The approach operates on point clouds, and assumes perfect knowledge of point correspondences. It doesn’t address the inherent properties of actual vision that make it difficult, namely occlusion in general and intra-object occlusion for deformable objects (especially cloth!) in particular. Segmentation and correspondence is mentioned under limitations. Assuming segmentation is fine given the current SotA. Perfect point-wise correspondence on the other hand is (a) ill-defined under deformation and (b) highly unrealistic in relevant real-world applications (think pile of clothes).
* Inferring a goal configuration for the object, as the paper does, might be one important step in a multi-stage (as opposed to end-to-end visuo-motor) system. But without showing how this helps in answering the tricky questions of e.g. initial gripper placement, disentangling the cloth, disambiguation of the cloth state (as a combined control and vision problem), dynamics, object-anchor interaction, etc., it is unclear whether this is really the right thing to do, or just a contrived problem without application in a complete physical robot setup.
* I find the experiments too weak to substantiate the bold claims of robustness and generalizability from the introduction. Experiments are limited to a single object type, a piece of cloth, and anchor, a hanger. The only property of the anchor that is varied is its pose. The size of the cloth, number (1 or 2), location and size of holes in the cloth, and the hanger pose are randomized. However, the choice of object and anchor geometry also greatly simplifies the task due to e.g. the hanger being largely rotationally symmetric and the only relevant part of the cloth being the hole, not the fabric around it.
* Evaluation is performed relative to various ablations only. No experimental comparison with other methods from the literature is done.
* No information on timing is provided. The number of (inverse) diffusion iterations, and the total computing time would have been critical to understand the applicability of the method on a real robot (operating under real-time constraints).
* If the task benchmark is explicitly stated as a contribution, why is no code/data provided for this?

**Minor Weaknesses**

* Links in the PDF don’t work, neither to the bibliography nor to the project website.
* There is no video in supplemental material. This would have been useful to e.g. visualize the diffusion process. It would have been critical to give readers a sense of the distribution of cloth and hanger configurations used at training and test time.
* The right-hand side of Fig. 1 is hard to read (too small, low contrast).

**Quality Of The Limitations Section:**

1

**Questions For Rebuttal:**

Please modify the paper to address the above weaknesses.

**Robotics Focus:**

2

**Summary Of Paper:**

The paper aims to address the problem of robotic manipulation for multimodal, deformable object placement tasks. The authors propose a deep neural network to predict a goal object configuration as dense per-point displacements given initial object and anchor point clouds. Deformation is achieved by per-point (as opposed to per-object) displacement. Multimodality is achieved by a diffusion transformer approach. Claimed contributions are the problem formulation, main method implementation, and a novel benchmark. The paper stops at providing a goal object configuration. A policy for task execution based on this is hinted at but not specified (neither in the paper nor the referenced Appendix C). Training data and experiments are based entirely on simulation. No physical robot (real or simulated) is involved. Experiments are done for a single task: hanging a piece of cloth on a hanger. Success rates are high (95-98%) for a fixed and medium (63-70%) for random cloth geometry.

**Summary Of Recommendation:**

I consider the lack of modeling of a physical robot (real or simulated) to be a blocker. The hard problems arising from the physics of the robot, the object and the anchor are not even mentioned. It is thus unclear whether the proposed formulation is a contrived problem or actually useful in a real application. The fact that the authors do not relate their approach to related work on deformable object manipulation and the limited experiments do not help with this either.

---

### Author Rebuttal · Authors · 2024-08-14

We upload an updated manuscript "TAX3D (Rebuttal).pdf", containing working links, various clarifications requested by the reviewers, as well as an additional experiment with a regression baseline supporting our method's diffusion-based approach for multimodal settings. Changes to the manuscript are highlighted for the reviewers' convenience. Please note that the updated manuscript pdf also contains the appendix.

We also upload a separate supplement "Additional Experiments.pdf" that contains details and visualizations of the additional experiments we performed for our method, both in simulation and the real world. Post-acceptance, we plan to incorporate information in "Additional Experiments.pdf" into the main paper/appendix. For now, we have separated them for the reviewers' convenience.

For specific responses to reviewer feedback, please see the official comments.

---

### Decision · Program_Chairs · 2024-09-04

**Decision:**

Accept

**Comment:**

Summarizing the strengths and weaknesses pointed out by the reviewers:

Strengths:
- The proposed method is an interesting and novel extension of prior work
- Well-written and easy to follow
- Proposed problem is a reasonable and good extension of prior problem settings

Weaknesses:
- All simulation experiments, not real world experiments
- Problem formulation is narrower than it needs to be
- Problem ignores intermediate states between initial and goal
- The way pointclouds are used is problematic, e.g., perfect clouds without occlusions are assumed
- Experiments don't include enough variation to justify generalization. Only evaluated on 1 task in simulation
- No comparisons with other baselines from the literature
- Not enough justification of the chosen architecture
- The description of novelty is missing
- The use of RSME as a metric may not yield the desired results

Overall, 2 reviewers recommend rejection (weak) and 1 acceptance (also weak). The reviewers brought up several valid criticisms that the authors need to address. The authors should carefully read each review and address the concerns as best as possible.

**Updated after rebuttal**: The authors' provided an extensive rebuttal, including changes to the paper and additional experiments. None of the reviewers replied after the rebuttal, so we must analyze the rebuttal to see if it adequately addressed the reviewers' concerns. Fixing the link to the project webpage was a major upgrade to the paper. It provides significant clarity to the methodology and results. From the videos, it is clear that not only are there real robot experiments, but that the method isn't simply predicting a goal pose, the target object is moved along a trajectory to hang it on the peg. Furthermore, the authors added an experiment hanging a bag in both sim and real, which doubles the number of tasks (previously there was only 1).

I find that the authors' changes during the rebuttal has significantly improved the paper and addressed most of the major concerns of the reviewers. Pre-rebuttal the reviewers were 1 weak accept and 2 weak rejects. However, with the new changes I believe this paper should be accepted *WITH ONE MAJOR CAVEAT*: The additional experiments in the second pdf of the rebuttal must be incorporated into the paper. Given the paper is already 3/4 page over the limit, it's likely many details will need to be moved to the appendix, but at least the results should be in the paper proper (never bury good results in an appendix).